# Investigating the quaternary structure of a homomultimeric catechol 1,2-dioxygenase: An integrative structural biology study

Arisbeth Guadalupe Almeida-Juarez[1]*, Shirish Chodankar[2], Liliana Pardo-López[3], Guadalupe Zavala-Padilla[4], Enrique Rudiño-Piñera[1]*

**1** Laboratorio de Bioquímica Estructural, Departamento de Medicina Molecular y Bioprocesos, Instituto de Biotecnología, Universidad Nacional Autónoma de México, Cuernavaca, Morelos, México, **2** National Synchrotron Light Source II, Brookhaven National Laboratory, Upton, New York, United States of America, **3** Laboratorio de Biotecnología Marina, Departamento de Microbiología Molecular, Instituto de Biotecnología, Universidad Nacional Autónoma de México, Cuernavaca, Morelos, México, **4** Unidad de Microscopía Electrónica, Instituto de Biotecnología, Universidad Nacional Autónoma de México, Cuernavaca, Morelos, México

\* enrique.rudino@ibt.unam.mx (ERP); arisbeth.almeida@ibt.unam.mx (AA)

## Abstract

The structural analysis of catechol 1,2 dioxygenase from *Stutzerimonas frequens* GOM2, SfC12DO, was conducted using various structural techniques. SEC-SAXS experiments revealed that SfC12DO, after lyophilization and reconstitution processes, can form multiple enzymatically active oligomers, including dimers, tetramers, and octamers. These findings differ from previous studies, which reported active dimers in homologous counterparts with available crystallographic structures, or trimers observed exclusively in solution for SfC12DO and its homologous isoA C12DO from *Acinetobacter radioresistens* under low ionic strength conditions. In some cases, tetramers were also reported, such as for the *Rodococcus erythropolis* C12DO. The combined results of Small-Angle X-ray Scattering, Dynamic Light Scattering, and Transmission Electron Microscopy experiments provided additional insights into these active oligomers' shape and molecular organization in an aqueous solution. These results highlight the oligomeric structural plasticity of SfC12DO, proving that it can exist in different oligomeric forms depending on the physicochemical characteristics of the solutions in which the experiments were performed. Remarkably, regardless of its oligomeric state, SfC12DO maintains its enzymatic activity even after prior lyophilization. All these characteristics make SfC12DO a putative candidate for bioremediation applications in polluted soils or waters.

## Introduction

Catechol 1,2-dioxygenases, C12DOs, constitute a class of non-heme iron-containing enzymes ($Fe^{+3}$) in the family of intradiol dioxygenases [1]. They play a crucial role in

**Data availability statement:** All data are within the manuscript or provided as Supporting Information files.

**Funding:** The investigation presented here was funded by the Dirección General de Asuntos del Personal Académico (DGAPA) at the Universidad Nacional Autónoma de México (UNAM) through PAPIIT grants IN226523 and IG200223. This research utilized resources from the 16-ID LIX beamline of the National Synchrotron Light Source II, a user facility managed by the U.S. Department of Energy (DOE) Office of Science, which the Brookhaven National Laboratory operates under Contract No. DESC0012704. AGA-J was supported by a doctoral scholarship (2019-000002-01NACF-12588) granted by CONAHCyT, Mexico, and by the incentive as a member of the program Ayudantes de Investigador nivel III provided by the SNII. ER-P gratefully acknowledges financial support from the institutional budget from Instituto de Biotecnología , UNAM, and the economic incentive from Sistema Nacional de Investigadoras e Investigadores (SNII), México.

**Competing interests:** The authors have declared that no competing interests exist.

the β-ketoadipate pathway, specifically in the cleavage of the aromatic ring of catechol (1,2-dihydroxybenzene), by introducing two oxygen atoms between carbons one and two of the aromatic ring [2]. This reaction leads to the formation of *cis-cis* muconate (*ccMA*), which subsequently enters the tricarboxylic acid cycle. Moreover, *ccMA* is a precursor of the adipic acid commonly used in the industry for the benzene-free synthesis of nylon 6,6 and other polymers [1,3,4,].

C12DOs have been identified in many organisms, spanning bacteria, fungi, and even higher plants [1,5–11]. Research on C12DOs has mainly focused on bacteria strains isolated from those substrates particularly affected by industrial residues, oil spills, and other contaminants [7]. Some bacteria capable of surviving and growing in these highly polluted environments have been isolated, for example, *Gordonia alkanivorans*, obtained from the oil-contaminated sludge of a local gas station in Taiwan [12]; *Paracoccus* sp. MKU1 isolated from the textile industrial effluent near Tirupur, India [13,14], and *Stutzerimonas frequens* GOM2 (*Pseudomonas stutzeri* GOM2) obtained from the oil-polluted substrate in the depth of southwestern Gulf of Mexico [15].

According to the crystallographic structures available in the PDB, native C12DOs are dimeric, as seen in PDB entries 2AZQ, 1DLM, 2XSR, 5UMH, 5TD3, 5VXT, and 3HGI. These dimers present a common hydrophobic zipper at the dimeric interface, with a pair of phospholipids embedded within the cavities of this interface [16]. However, despite the dimeric homogeneity observed in crystallographic structures, some C12DOs present different oligomeric states in solution. For example, certain C12DOs exist as monomers, such as C12DO from *P. aeruginosa* TKU002 ($M_W$ 22 kDa) [17], or trimers, such as the C12DO from *Trichosporon cutaneum* WY 2–2 ($M_W$ 105 kDa) [18], and the C12DO from *Paracoccus* sp. MKU1-A ($M_W$ 121.4 kDa) [14].

In some cases, C12DOs enzymes also show a flexible quaternary structure. For example, C12DO from *Rhodococcus erythropolis* AN-13 is mainly described as monomeric, but in the absence of salt in the buffer, oligomerizes into stable tetramers ($M_W$ 150 kDa) [19]. The Isoform A of C12DO from *Acinetobacter radioresistens,* can exist in two active forms: as a trimer ($M_W$ 112.4 kDa) and as a dimer ($M_W$ 77.6 kDa), depending on the ionic strength conditions [20]. For catechol 1,2-dioxygenase from *S. frequens* GOM2, SfC12DO, previously named PsC12DO, a transition from trimers into dimers was described when the buffer's salt concentration mimics seawater [15].

This intriguing behavior of oligomer transitions in some C12DOs implies that the biochemical environment, particularly the ionic strength, seems to be the critical factor driving those changes [15,20]. Furthermore, these structural alterations may also be associated with variations in its enzymatic activity and the protein aggregation-dissociation behavior [15,19–22].

In this study, we utilized a combination of analytical techniques, including Small-Angle X-ray Scattering (SAXS), Size Exclusion Chromatography (SEC), Dynamic Light Scattering (DLS), and Transmission Electron Microscopy (TEM) to gain fundamental insights into the quaternary structure behavior of SfC12DO in solution. By understanding how structural changes influence enzymatic activity, we can

better understand the potential application of these enzymes in bioremediation processes, including those involving seawater affected by oil spills. This work represents the first application of SAXS to SfC12DO, revealing new low-resolution structural characteristics of this catechol 1,2-dioxygenase, which exhibits several active oligomeric states in solution.

## Materials and methods

### Protein expression and purification

To express SfC12DO we employed *E. coli* BL21 (DE3) cells harboring the pETCDOPs vector, which contains the CatA gene responsible for encoding catechol 1,2-dioxygenase derived from *S. frequens* GOM2. The details of the vector construction process are described in [15].

A pre-culture of *E. coli* BL21 (DE3) cells transformed with pETCDOPs was inoculated into 1L of LB medium containing 100 µg/mL ampicillin and grown at 37 °C, 200 rpm. Once the cell culture reached an $OD_{600}$ of 0.6–0.7, protein expression was induced by adding IPTG at a final concentration of 1 mM and further incubated for 5 h. After this incubation, the cell culture was harvested by centrifugation at 4,600 $g$ at 4 °C. The cells were disrupted by sonication at 37% amplitude and 40 s ON/ 40 s OFF cycle for 30 min (Sonics Vibra-cell Ultrasonic Processor), using *lysis Buffer* (S1 Table) throughout the sonication and purification processes. Affinity chromatography was conducted following three gradient steps of Imidazole (5 mM, 20 mM, and 500 mM). The SfC12DO purity was confirmed by SDS-PAGE (Sodium Dodecyl Sulfate - Polyacrylamide Gel Electrophoresis).

### Sample preparation

The purified SfC12DO was concentrated and filtered using Amicon® Ultra-15 centrifugal filters with a molecular weight cutoff of 30 kDa (Merck, Germany), undergoing three changes of 10 ml of MilliQ water at 4 °C.

Enzyme aliquots containing 5 and 10 mg were frozen at -80 °C and lyophilized for 12 hours using a LABCONCO Freezone® Legacy system. The lyophilized samples were then stored at -20 °C. Before analysis, the lyophilized SfC12DO samples were reconstituted by adding Buffer B (S1 Table) to achieve the desired concentration for each experiment performed in this work.

### Circular Dichroism spectra (CD)

CD spectra of SfC12DO were acquired using a Jasco J-715 CD Spectrometer (JASCO Analytical Instruments) to assess the sample's secondary structure composition. SfC12DO samples were resuspended in degassed and filtered Buffer C (S1 Table) and the protein concentration was adjusted to 0.3 mg/ml, with a sample volume of 200 µl. Measurements were performed in triplicate using a 1 mm pathlength cell over a wavelength range between 190 and 260 nm at 25 °C, controlled by a Peltier temperature cell holder (PTC-4235; JASCO). The ellipticity was reported as mean residue ellipticity ([θ]mre, in deg cm² dmol⁻¹), and the data were analyzed using the BestSel web server (http://bestsel.elte.hu).

### Enzyme activity assay

The specific activity of SfC12DO was determined spectrophotometrically using a Cary 60 UV-vis spectrophotometer (Agilent Technologies). The enzymatic reaction was conducted in Buffer B (S1 Table) and monitored by measuring the formation of product (*ccMA*) at 260 nm ($\varepsilon$ = 16.8 M-1 cm-1) [23] at 40 °C for 1 min. One unit (U) of the enzyme was defined as the amount of the enzyme required to catalyze the formation of 1 µmol of ccMA per min.

### Size Exclusion Chromatography (SEC)

For SEC analysis, an aliquot of SfC12DO of 500 µl volume at 10 mg/mL was centrifuged at 12,800 $g$ for 5 min to eliminate aggregates. Following centrifugation, the sample was subjected to SEC using a Sephacryl 200 HiPrep 16/60 column on

an ÄKTA Prime - FPLC System. The column was pre-equilibrated with Buffer B (S1 Table). The purification process was conducted at a 1.0 mL/min flow rate, and the absorbance at 280 nm was monitored. The molecular mass was confirmed by comparing the retention time to the calibration curve standards. Lysozyme (14 kDa), Horseradish (44 kDa), PADA-I* (52 kDa), BSA (66 kDa), and ɣ-globulins (150 kDa). *PADA-I was kindly provided by Prof. Marcela Ayala´s Group, IBt, UNAM. All other proteins used for column calibration were sourced from Sigma-Aldrich.

### Denaturing and Non-denaturing PAGE

*SDS-PAGE* was carried out by the denaturing Tris-Gly-SDS buffer system (Laemmli, 1970). A Precision Plus Protein™ prestained marker (Bio-Rad) was used as the standard (in the range from 10- to 250 kDa). Electrophoresis was performed at 120 V at room temperature, and gels were stained with Coomassie blue.

*SEMI-NATIVE-PAGE* was carried out using the Tris-Gly-SDS buffer system (Laemmli, 1970) without adding detergent or denaturing agents. BSA was used as a protein size standard. Electrophoresis was performed at 100 V at 18 °C, and gels were stained with Coomassie blue.

### Dynamic Light Scattering analysis (DLS)

DLS analysis was performed using the Zetasizer Nano Z system (Malvern Instruments) on a SfC12DO protein sample solution in Buffer B (S1 Table) with a volume of 1 mL at 1 mg/mL. The measurements were carried out in a quartz cuvette, and the scattered light was collected at a fixed angle of 173 °. The experiments were performed in triplicate to ensure accuracy and reliability. The temperature during the analysis was maintained at 40 °C.

### Transmission Electron Microscopy (TEM)

SfC12DO sample solutions with concentrations in a range between 0.3 and 0.5 mg/mL were negatively stained using the methodology described in [24]. Samples were applied onto EMS carbon copper grids and washed briefly in sterile water for 5 s. Uranyl acetate (1% w/v) (Electron Microscopy Science) was used on the grid for 30 s, and the excess stain was blotted directly. Subsequently, the uranyl acetate application was left to dry for 5 min before being mounted in the sample holder. TEM was performed at 80 kV using a ZEISS LIBRA 120 Transmission Electron Microscope. Images were recorded with a GATAN CCD, and the data obtained was analyzed with the Digital Micrograph Software (Gatan, Pleasanton, CA).

### Structural modeling of SfC12DO

To obtain tridimensional models of SfC12DO based on its amino acid sequence, we used AlphaFold2 [25]. Multimer analysis using the AlphaFold2-multimer was applied to model ten putative SfC12DO oligomers (monomer, dimer, and trimer) that were predicted using ColabFold v1.5.5 (https://colab.research.google.com/github/sokrypton/ColabFold/blob/main/AlphaFold2.ipynb) [26] and tetramer to decamer models were generated through the COSMIC$^2$ platform (https://cosmic2.sdsc.edu) [27]. For each modeling run, five structures were generated and selected based on the top predicted local-distance difference test values (plDDT).

### Small-Angle X-ray Scattering (SAXS)

SAXS experiments were performed at the LIX-beamline (16-ID) of the National Synchrotron Light Source II (NSLS-II, Upton, NY). The SfC12DO sample was previously resuspended in Buffer B (S1 Table) and centrifuged. SAXS data was analyzed using two different methods. For *static mode,* the buffer scattering was determined by subtracting the scattering of the empty cuvette from that of the degassed Buffer B (S1 Table). The protein and buffer scattering measurements were repeated in three separate experimental sessions, and the results were averaged. The data were processed by the ATSAS version 3.0.5–2 package [28,29]. The data quality is assessed using AUTORG, and SAXS profiles were further analyzed with OLIGOMERS to interpret multi-component mixtures [30].

 

For SEC-SAXS (size exclusion chromatography coupled to small angle X-ray scattering) analysis, the sample was applied into a pre-equilibrated Superdex 200 5/150 column (GE Healthcare) with degassed Buffer B (S1 Table) as the mobile phase. The eluent was directly coupled to X-ray scattering data collection.

Data collection utilized a wavelength of 0.819 Å and a scattering angle of $0.006 < q < 3.0$ Å$^{-1}$, with an exposure time of 2 s. Scattering data was collected using Pilatus 3X 1M and 900 K detectors positioned at 3.7 m and 300 m sample-detector distances, respectively.

## SAXS data processing

Initial data processing was performed using LiXTools (https://github.com/NSLS-II-LIX/lixtools) and py4xs (https://github.com/NSLS-II-LIX/py4xs). The determinations of I (0), molecular weight (MW) (Bayesian inference method), maximum dimension particle (Dmax), and radius of gyration (Rg) were calculated using PRIMUS in the ATSAS version 3.0.5–2 package [28]. Rg was calculated from the slope of the Guinier plot, described as ln(I) vs. $q^2$, where $q = 4 \pi \sin (\theta) / \lambda$ is the scattering vector (2θ is the scattering angle and λ is the wavelength).

For *ab initio* modeling, WAXSiS [31] (https://waxsis.uni-saarland.de) was used to fit the SAXS data, considering the explicit solvent modeling of protein [31,32]. The SfC12DO models were initially generated using AlphaFold2 and subsequently refined with SREFLEX [33]. Following these steps, for single component mixture, the quality of experimental data related to the protein model and hydration shell fitting was assessed using CRYSOL [34] and FoXS (https://modbase.compbio.ucsf.edu/foxs/) [35,36]. This methodology for SAXS processing data is detailed in Fig 1.

## Results and discussion

In a preliminary characterization of SfC12DO oligomers, freshly purified SfC12DO was subjected to an initial SAXS (static mode) measurement. However, the samples exhibited signs of protein unfolding (S2 Fig.). During shipping to the synchrotron facility, the protein was unable to withstand the transport conditions and precipitated. To mitigate aggregation and enhance stability during shipping and long-term storage, we investigated lyophilization as a preservation strategy.

### Secondary structure analysis

Secondary structure analysis of lyophilized SfC12DO was crucial to determine whether the protein remained properly folded. The predicted secondary structure of SfC12DO can be visualized (Fig 2A). CD analysis of SfC12DO showed slight changes in the content of secondary structure elements between non-lyophilized and lyophilized SfC12DO. The spectra reveal similar profiles with slight variations, particularly in the α-helix characteristic regions (~208 nm and ~222 nm) (Fig 2B). The distribution of α-helices and β-sheets closely resembled the typical C12DO secondary structure, showing significant similarity to Iso A (C12DO) from *A. radioresistens* S13 [2], which shares 53% sequence identity with SfC12DO, and C12DO from *Achromobacter xylosoxidans* DN002, which has 50% sequence identity with SfC12DO [37] (Fig 2C). Non-lyophilized SfC12DO contains 22.8% α-helices and 27.5% β-sheets, while lyophilization and reconstitution result in slight increases in α-helices (29.2%) and decreases in β-sheets (26.2%).

A comparison of the predicted SfC12DO model with other C12DO models, including those from *A. xylosoxidans* DN002 and the IsoA variant of *A. radioresistens*, as well as the experimental structure of IsoB from *A. radioresistens* (PDB entry 2XSR), demonstrates that they all share the same overall folding pattern, particularly in the N-terminal and C-terminal regions (Fig. 2D). Additionally, the catalytic site is well conserved across all C12DO models.

### SEC analysis and semi-native PAGE

The SDS-PAGE analysis of SfC12DO (lyophilized and non-lyophilized) reveals an intense band near 35 kDa (Fig 3A), corresponding to monomeric subunit of SfC12DO in denaturing conditions. In the lyophilized sample a variety of bands above 55 kDa are observed, indicating the presence of higher-order species. Similarly, the semi-native-PAGE analysis of

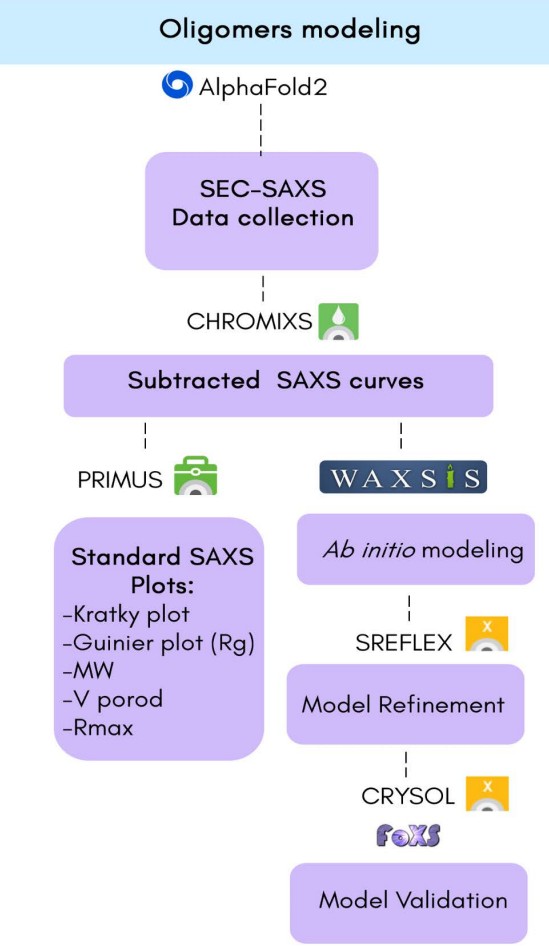

**Fig 1. Schematic representation of SAXS data analysis process.**

the lyophilized SfC12DO shows the presence of various oligomeric forms, whereas in the non-lyophilized SfC12DO, only a dimer of approximately 70 kDa was detected (Fig 3B).

Previous studies have reported the existence of various oligomers in the lyophilized C12DOs from *P. putida* [38]; however, a detailed characterization of these oligomers is still needed to understand their structural and functional implications. The observed conformational differences are likely attributable to artifacts caused by the lyophilization process [38], which may promote non-specific aggregation or stabilize transient oligomeric forms that do not necessarily reflect the solution-state behavior of SfC12DO. The presence of distinct putative oligomers regarding the dimeric state was also confirmed by SAXS (*static mode*) OLIGOMERS (S2 Fig).

To separate the different SfC12DO oligomers, a size-exclusion chromatography (SEC) analysis was performed (Fig 4). The chromatogram corresponding to the lyophilized SfC12DO revealed four peaks at different retention volumes: p1 (46.5 ml), p2 (49 ml), p3 (54 ml), and p4 (61 ml). In contrast, for non-lyophilized SfC12DO, only a single peak corresponding to the dimeric state was observed (red dashed line, 61.21 ml). Calibration with protein standards enabled the estimation of molecular weights ($M_w$) for peaks 3 and 4 (p3 and p4). However, the initial peak (p1) contained aggregates with molecular weights above exceeding the resolution limits of the SEC column employed, and the oligomers presented in p2 were bigger than the protein standard calibrators used in this experiment. The p3 fraction corresponds to an oligomer

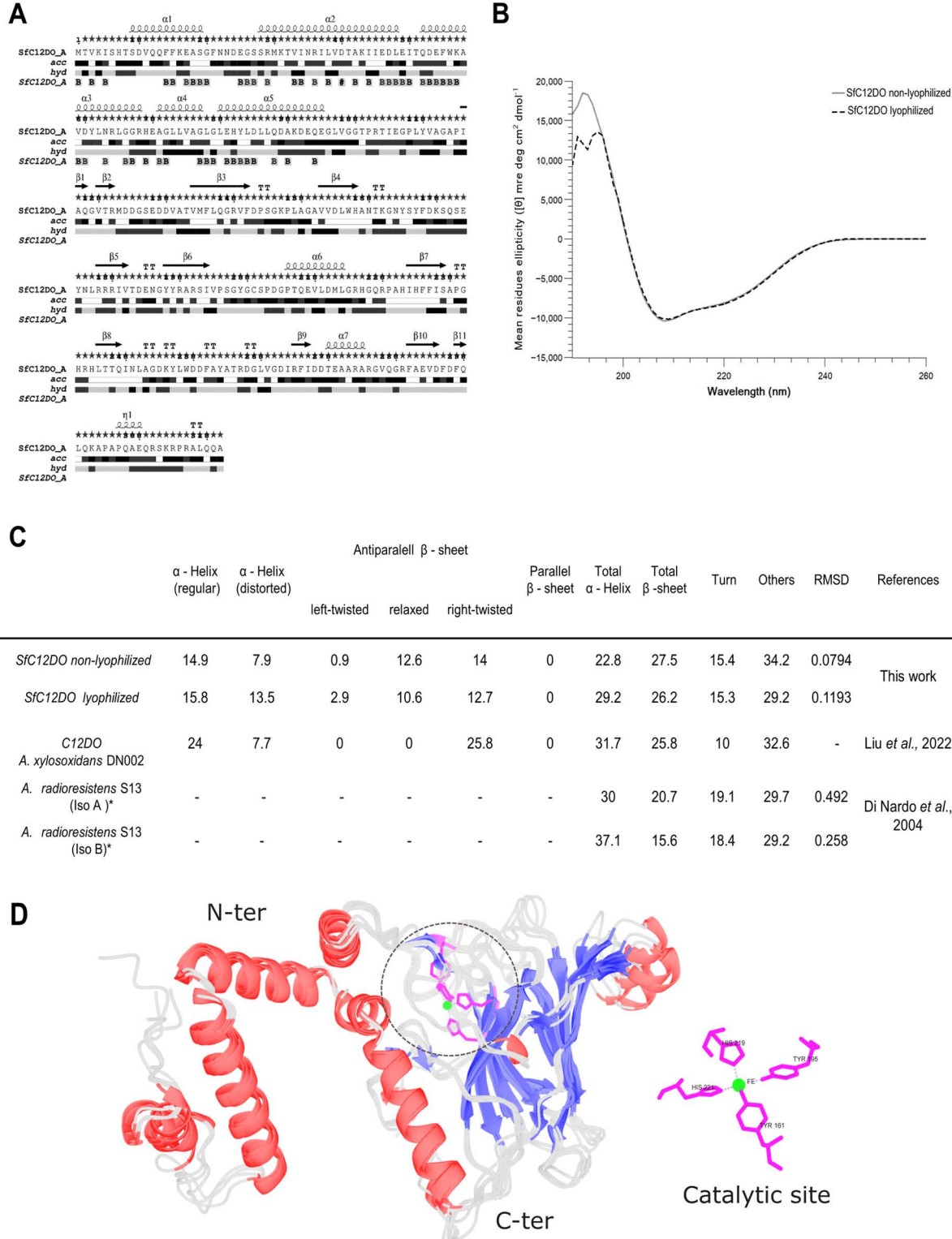

**Fig 2. Secondary Structure of SfC12DO. A)** The secondary structure of SfC12DO was visualized using ESPript 2.0. In this panel, "acc" represents the solvent accessibility index, while "hyd" stands for the hydrophobicity index. **B)** Circular dichroism (CD) spectra illustrate the secondary structure content in SfC12DO. The solid gray line represents the non-lyophilized SfC12DO, whereas the dashed black line represents the lyophilized SfC12DO. **C)** The

| | α - Helix (regular) | α - Helix (distorted) | Antiparalell β - sheet | | | Parallel β - sheet | Total α - Helix | Total β -sheet | Turn | Others | RMSD | References |
|---|---|---|---|---|---|---|---|---|---|---|---|---|
| | | | left-twisted | relaxed | right-twisted | | | | | | | |
| *SfC12DO non-lyophilized* | 14.9 | 7.9 | 0.9 | 12.6 | 14 | 0 | 22.8 | 27.5 | 15.4 | 34.2 | 0.0794 | This work |
| *SfC12DO lyophilized* | 15.8 | 13.5 | 2.9 | 10.6 | 12.7 | 0 | 29.2 | 26.2 | 15.3 | 29.2 | 0.1193 | |
| *C12DO* *A. xylosoxidans* DN002 | 24 | 7.7 | 0 | 0 | 25.8 | 0 | 31.7 | 25.8 | 10 | 32.6 | - | Liu *et al.,* 2022 |
| *A. radioresistens* S13 (Iso A )* | - | - | - | - | - | - | 30 | 20.7 | 19.1 | 29.7 | 0.492 | Di Nardo *et al.,* 2004 |
| *A. radioresistens* S13 (Iso B)* | - | - | - | - | - | - | 37.1 | 15.6 | 18.4 | 29.2 | 0.258 | |

accompanying table compares the content of secondary structure elements—α-helices, turns, coils, and β-sheets—between samples. The root mean squared deviation (RMSD) is provided, comparing the calculated and experimental CD spectra. **D)** The superposition of different models (denoted as C12DOs) is shown. The models for SfC12DO, C12DO from *A. xylosoxidans* DN002, and *A. radioresistens* S13 (IsoA) were predicted using AlphaFold2. Meanwhile, the structure of C12DO from *A. radioresistens* (Iso B) was obtained from the PDB entry 2XSR. The domains are color-coded by secondary structure: red for α-helices, blue for β-sheets, and gray for loops, with the catalytic site highlighted in magenta.

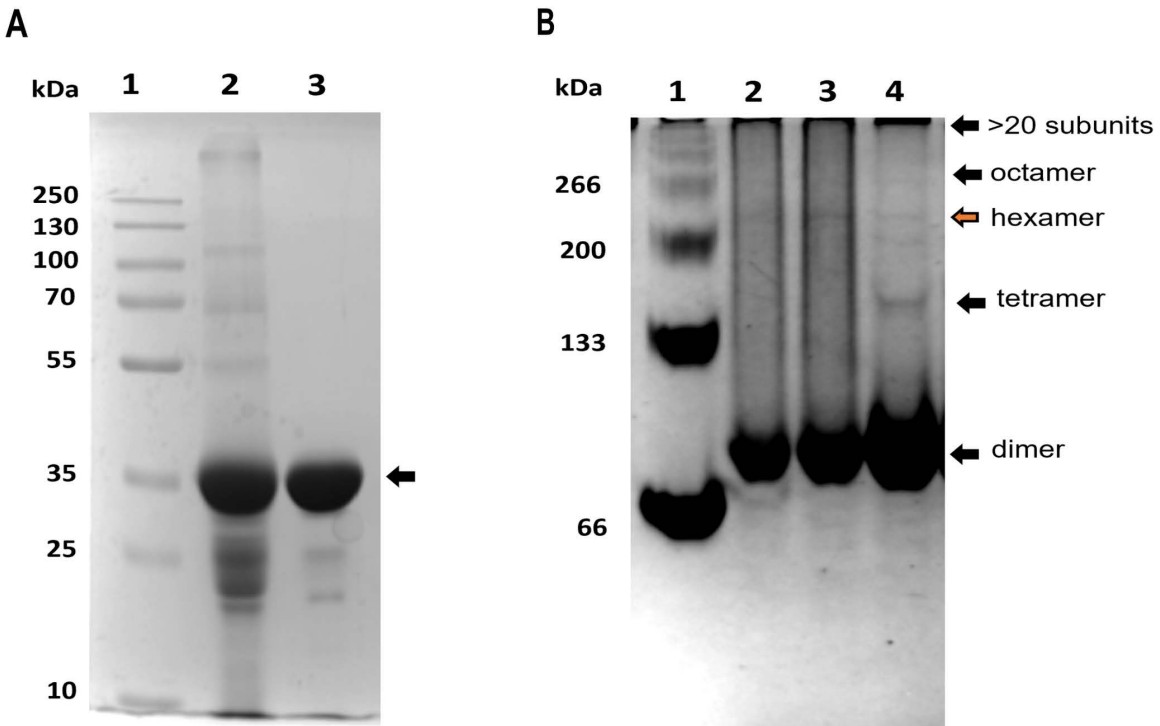

**Fig 3. Oligomerization Analysis of the SfC12DO Protein. A)** SDS-PAGE (12%) was performed on the purified SfC12DO to compare lyophilized and non-lyophilized samples**.** Lane 1 contains a molecular weight marker (kDa). Lane 2 shows the lyophilized SfC12DO, while Lane 3 displays the non--lyophilized SfC12DO. **B)** A semi-native PAGE (8%) was conducted to analyze the oligomeric distribution of SfC12DO. Lane 1 includes BSA as a molecular marker (kDa). Lanes 2 and 3 are two replicates of the non-lyophilized SfC12DO, and Lane 4 presents the lyophilized SfC12DO. The semi-native PAGE analysis of the lyophilized SfC12DO reveals various oligomeric species, including dimers, tetramers, hexamers, octamers, and high-ordered aggregates (greater than 20 subunits).

of approximately 113 kDa, and the p4 with a molecular weight of approximately 77 kDa overlapping with the dimeric state found in the non-lyophilized SfC12DO and also resembled those $M_w$ observed for the dimer in the semi-native-PAGE (Fig 3). A comparative analysis with protein standards unveiled the presence of three new distinct larger oligomers, corresponding to peaks p1, p2, and p3, the three surpassing the molecular weight of the trimeric (~108 kDa) and the dimeric (~80 kDa) SfC12DO reported in [15].

## SfC12DO activity assay

The enzymatic activity of four different peaks (p1, p2, p3, and p4) was evaluated using catechol as a substrate (Fig 5). Despite being lyophilized, peaks p3 (16.49 ± 1.2 U/mg) and p4 (15.42 ± 0.26) exhibited comparable activity to the non-lyophilized protein (12.44 ± 0.61 U/mg). This preservation of activity in C12DOs after lyophilization has also been observed in C12DO from *P. putida* DSM 437, where the lyophilized C12DO maintained its catalytic

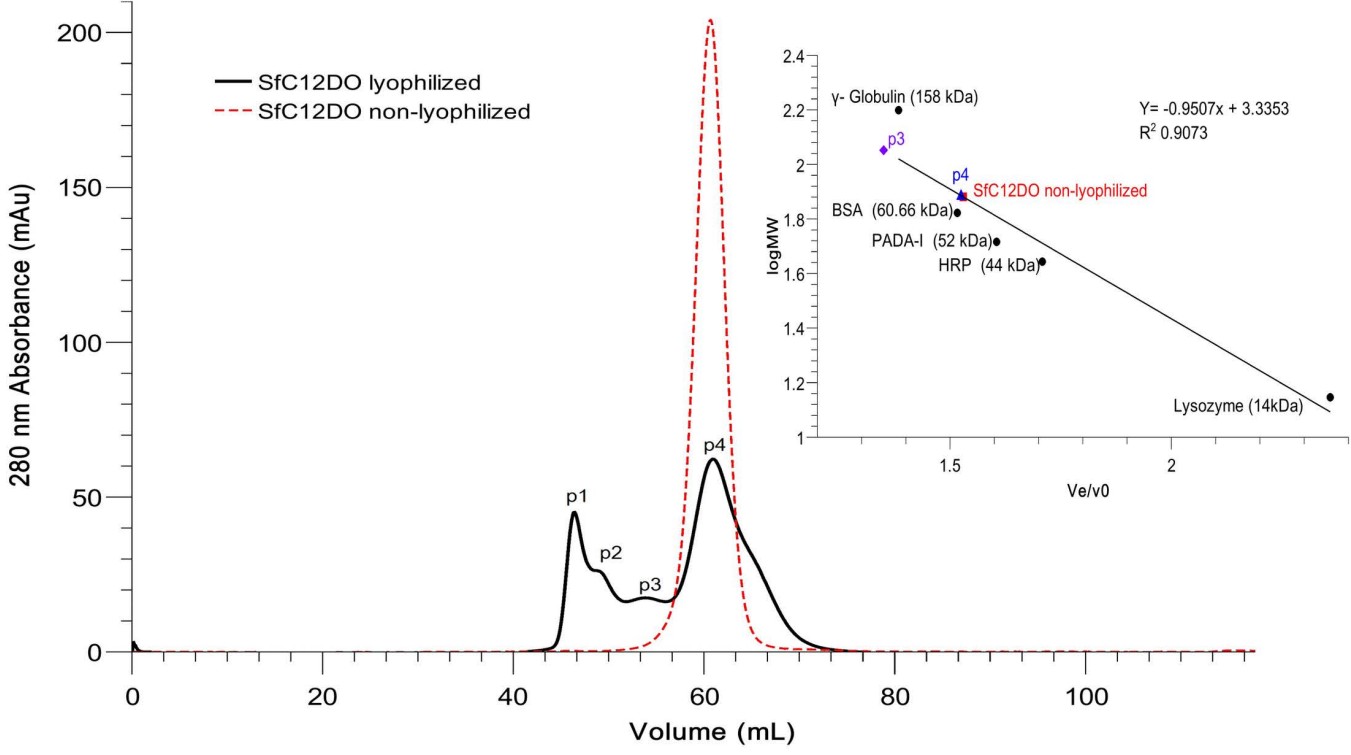

**Fig 4. SEC elution profile of SfC12DO (lyophilized and non-lyophilized) SfC12DO was eluted from a Sephacryl 200 HiPrep 16/60 column.** Four distinct peaks were observed for SfC12DO lyophilized: peak 1 (p1) and peak 2 (p2) with undetermined molecular weights, peak 3 (p3) with an estimated MW of ~113 kDa, and peak 4 (p4) with a MW of ~77 kDa. The red dashed line represents the elution peak of non-lyophilized dimeric SfC12DO, also at MW of ~77 kDa. On the right, the calibration curve standard is shown.

function in the presence of *n*-hexane [38]. However, larger aggregates (p1 and p2) exhibited a significant reduction in specific activity, with a decrease of 77.89% (2.75 ± 1.8 U/mg) and 44.13% (5.49 ± 0.58 U/mg), respectively, compared to the non-lyophilized SfC12DO dimeric sample. The study of enzymatic catalysis in non-conventional media is gaining increasing biotechnological interest, as these conditions can significantly affect enzyme properties [38]. In the case of SfC12DO, lyophilization has been observed to influence both its oligomeric state and enzymatic activity.

## SEC - SAXS analysis

The four peaks obtained in SEC were subjected to SEC-SAXS data collection to further analyze the oligomerization of SfC12DO in solution. This revealed the structural organization and size of p1, p2, p3, and p4.

The molecular weights of each oligomer were determined using Bayesian Inference (PRIMUS). The calculated molecular weights of p1 (833.40 kDa), p2 (318.45 kDa), p3 (157.05 kDa), and p4 (72.40 kDa) were utilized to classify the oligomeric states present in the SfC12DO sample. Given that the monomeric unit of SfC12DO has a molecular weight of 35.76 kDa, it can be deduced that p1 corresponds to aggregates with more than 20 subunits, p2 is an octamer, p3 is a tetramer, and p4 is a dimer. Differences between the apparent molecular weights of oligomers analyzed by SEC and the theoretical molecular weight of the SfC12DO monomer (35.76 kDa) may be due to structural modifications in SfC12DO's secondary structure or, more likely, to flexibility in certain protein regions as indicated by the Kratky plots and CD spectra analyses. Table 1 contains all experimental SAXS parameters.

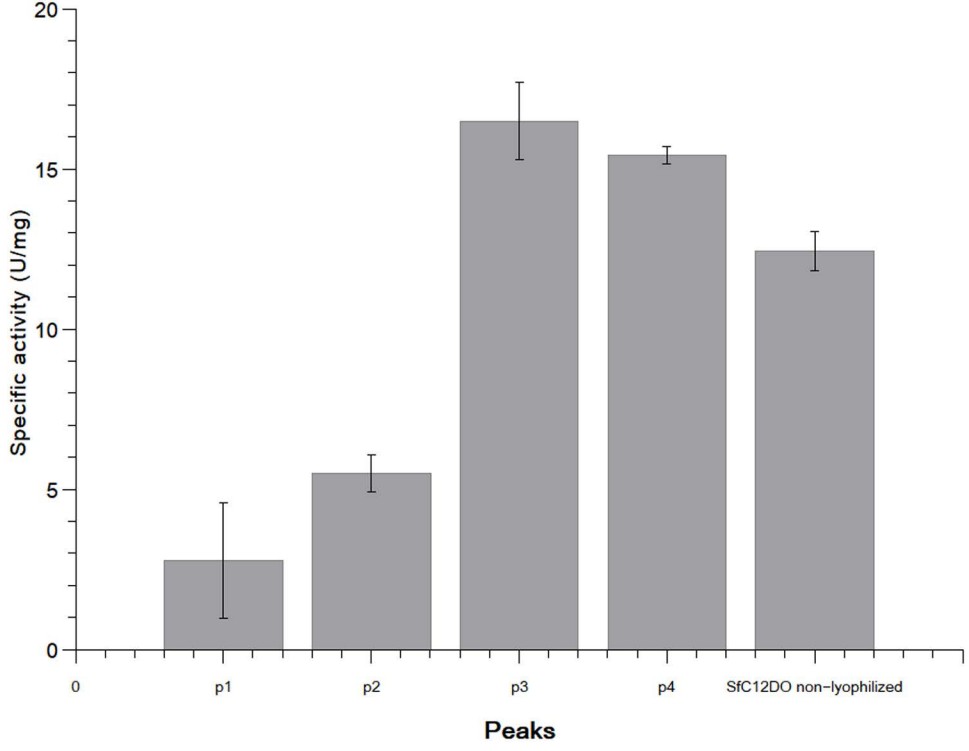

**Fig 5. Specific enzymatic activity of SfC12DO (U/mg).** The specific activity was monitored by measuring the formation of ccMA at 260 nm ($\varepsilon = 16.8$ M-1 cm-1) at 40 °C for 1 min. One unit (U) of the enzyme was defined as the amount of the enzyme required to catalyze the formation of 1 μmol of ccMA per min. The specific activities obtained were: 1: p1 (2.75±1.8 U/mg), 2: p2 (5.49±0.58 U/mg), 3: p3 (16.49±1.2 U/mg), 4: p4 (15.42±0.26), and non-lyophilized SfC12DO. 5: SfC12DO non-lyophilized (12.44±0.61 U/mg).

Discrepancies were observed between the Rg values determined by ATSAS and WAXSiS analyses, possibly due to the hydration shell considered in the WAXSiS calculation. The Rg values for the four different oligomers were calculated: for ATSAS, p1 (83.39 Å), p2 (59.59 Å), p3 (45.16 Å), and p4 (32.10 Å), while for WAXSiS, the Rg values were p1 (not calculated because of the lack of a three-dimensional model), p2 (48.79 Å), p3 (35.97 Å), and p4 (31.92 Å). Guinier plots for all oligomers correlated well with the Rg values obtained through the distance distribution function, P(r), analysis (S4 Fig). P(r) results indicated Rmax range values from 110 to 247 Å. The Kratky plot displayed a semi-bell-shaped peak, indicative of a partially intrinsically disordered structure in SfC12DO (S5 Fig).

The three-dimensional models of the three oligomers of SfC12DO, generated using WAXSiS based on SAXS data, reveal a predominantly globular shape (Fig 6). The experimental SAXS data indicate that the dimer adopts a more extended conformation than that observed in both, the crystallographic structures of homologs deposited in the PDB (Entries: 2AZQ, 1DLM, 2XSR, 5UMH, 5TD3, 5VXT, and 3HGI**),** and the models predicted by AlphaFold2. When scoring 10,000 conformers against SAXS datasets with CRYSOL, the best-fitting structure clusters displayed an elongated N-terminus for the dimeric SfC12DO SAXS three-dimensional model.

The obtained three-dimensional model for SfC12DO dimer indicates that the SAXS dimer model has a wider open angle between the N-terminal and other regions labeled in this work as region I (residues 1–28), region II (residues 197–222), region III (residues 268–282), and region IV (residues 293–312) in the solution state compared to its crystallographic counterparts (Fig 7). These differences were previously named "variability regions" in SfC12DO [15]. The root-mean-square deviations (RMSD) analysis showing these analyses is presented in Fig 7C.

**Table 1. SAXS data collection and analysis parameters.**

| Data collection parameters | | | | |
|---|---|---|---|---|
| Beamline | 16-ID LIX, NSLS-II | | | |
| Detector | Pilatus 3X 1M and 900K | | | |
| Beam size (mm) | 0.5 x 0.5 | | | |
| Energy (keV) | 13.5 | | | |
| Wavelength (Å) | 0.819 | | | |
| Sample- to- detector distance (m) | 3.7 | | | |
| q range (Å -1) | $0.006 < q < 3.0$ | | | |
| Exposure time (s) | 2 | | | |
| Temperature (K) | 293 | | | |
| Data collection mode | | | | |
| Structural parameters | Dimer (p4) | Tetramer (p3) | Octamer (p2) | >20-mer (p1) |
| Guinier analysis I (q=0) (Å) | 37.9±0.054 | 22.49±0.10 | 23.20± 0.28 | 37.9±0.054 |
| Rg (Å) (ATSAS) | 32.10±0.07 | 45.16±0.29 | 59.59±0.94 | 83.39±1.43 |
| Rg (Å) (WAXSIS) | 31.9225 | 35.9708 | 48.799 | – |
| MW (kDa) (ATSAS) | 72.4 | 157.05 | 318.45 | 833.476 |
| Rmax (Å) | 110.04 | 158.67 | 183.5 | 247.66 |
| Porod volume (A-1) (ratio V p/ calculated M) | 100251 (1.38) | 226669 (1.44) | 620131 (1.95) | 1872500 (2.24) |
| Molecular mass determination | | | | |
| Molecular mass Mr from I (0) in P(r) (kDa) | 72.4 | 157.05 | 318.45 | 833.476 |
| Calculated monomeric Mr, from sequence (kDa) | 35.76 | | | |
| WAXSIS MODEL parameters | | | | |
| X2 (total estimate from WAXSIS) | 3.64598±0.45 | 4.28279±0.094 | 1.75532±0.064 | Unmodeled |
| Software employed | | | | |
| Primary data reduction | PRIMUS | | | |
| Data processing | GNOM | | | |
| *Ab initio* analysis | WAXSIS | | | |
| Model comparison | FoXS | | | |
| Tertiary structure modeling | AlphaFold2 | | | |
| Computation of model intensities | Crysol | | | |
| Three-dimensional graphics representations | PyMol | | | |

These differences could arise from the inherent flexibility of these regions in solution, allowing them to take on multiple conformations as observed in SAXS, but being restricted in their crystallized form. Such variations emphasize the importance of using both methods to fully comprehend the different shapes a C12DO can adopt [39]. This may also explain how SfC12DO or certain C12DO oligomerizes, as they can take various structural arrangements without affecting their enzymatic activity.

The SAXS model of SfC12DO in tetramer and octamer form reveals previously undescribed conformations for C12DOs (Fig 8). These two SAXS conformations are likely possible and active in solution. In contrast, substrate diffusion problems might prevent it from efficiently reaching all interaction target sites due to its size, negatively affecting the enzymatic activity in the octameric structure. Given the differences between X-ray crystallography and SAXS techniques in terms of structural detail and experimental conditions, the results, while distinct, indicate that the crystallographic dimer (like the dimer presented in the PDB entry 2AZQ) is present within the tetramer (Fig 8A). In the octamer, an arrangement of different SfC12DOs dimeric conformations is observed (Fig 8B).

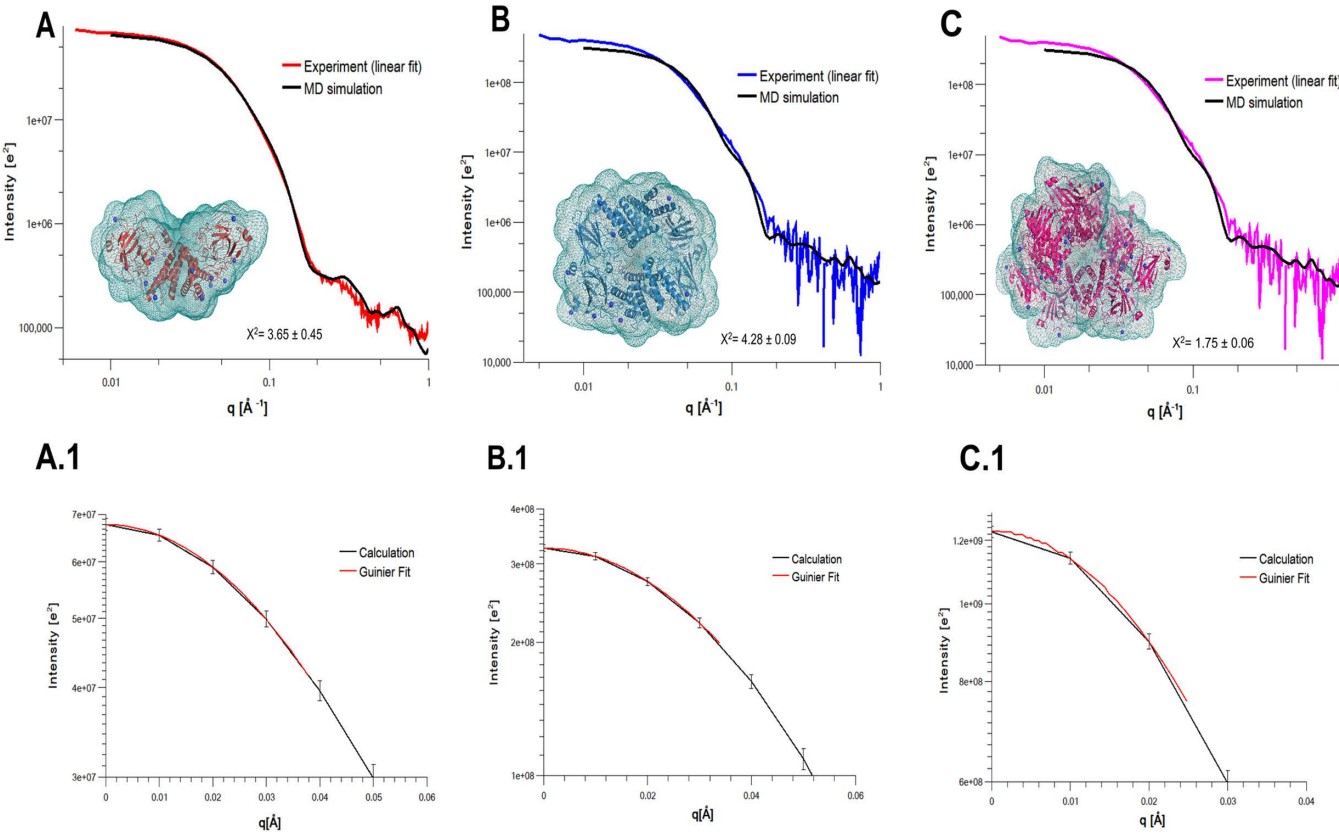

**Fig 6. SAXS intensity profiles of SfC12DO oligomers.** Processed solution scattering intensity patterns of oligomers, p2, p3, and p4 are shown. Each panel corresponds to a different oligomeric state: **A)** Dimer (p4), **B)** Tetramer (p3), **C)** Octamer (p2). The scattering profiles are plotted as intensity versus scattering vector **(q)**, with the respective structural envelopes obtained through SAXS modeling. The Guinier plots for each oligomer are included (A.1, B.1, and C.1), confirming the quality of the SAXS data and the absence of significant interparticle interactions.

## Characterization of oligomers by TEM and DLS

DLS initially evaluated the protein aggregation, and the calculated hydrodynamic diameters were between 7 and 20 nm. Subsequently, the determination of higher oligomers was confirmed by Transmission Electron Microscopy (TEM) (Fig 9). The resulting images using negatively stained contrast particles of different sizes correspond to the oligomeric species eluted in the SEC experiments (sizes from 75 kDa to 800 kDa). Due to the considerable variation in size, a detailed analysis using single-particle methods was not attempted. Nevertheless, a comparative study of TEM particles allowed for estimating their diameter. The TEM estimated particle diameter was consistent with the results obtained from SAXS.

The SAXS-derived structural parameters mirror the SEC, native electrophoresis, TEM, and DLS results, consistent with different enzymatically active oligomeric forms of SfC12DO described in this work. These are highly unusual findings, given that other C12DOs have yet to be studied by SAXS and other techniques that evaluate their oligomeric behavior in solution at the level described in this work. Our results suggest that our SfC12DO preparations are composed of various functional oligomers that retain, at different levels, their functional activity in solution.

In this study, the dimeric form was the only SfC12DO oligomeric state consistently present across all the experimental approaches analyzed. This contrasts with the trimeric form, proposed to appear only under certain optimal conditions (low ionic strength conditions) [14–16,20,40]. Interestingly, the trimeric SfC12DO form did not appear after

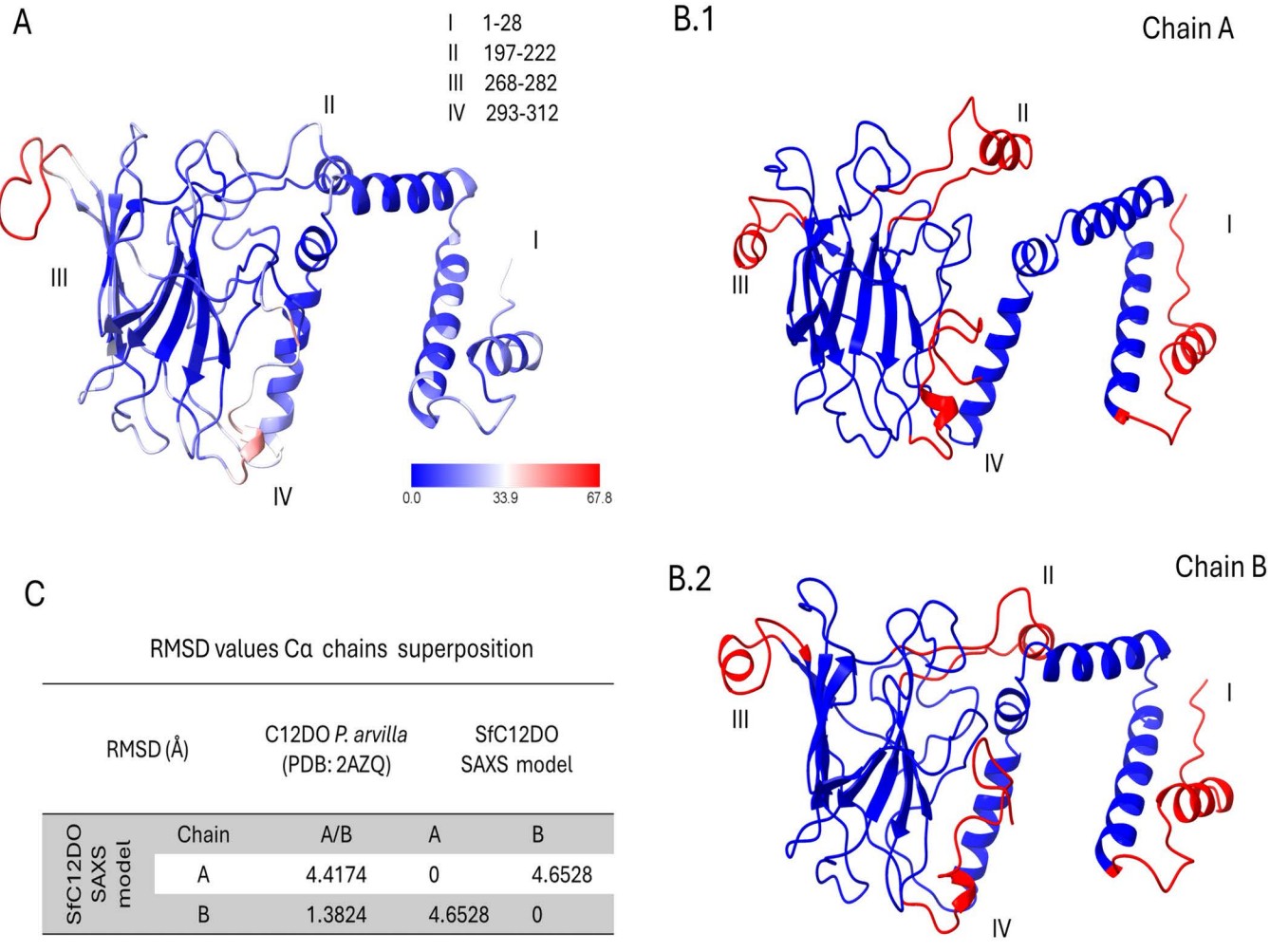

**Fig 7. Comparative analysis of homologous C12DO structure with SfC12DO SAXS dimeric model structure and flexible regions. A)** The three-dimensional structure of the homologous C12DO (PDB: 2AZQ) from *P. putida* shows that regions III and IV are flexible (colored in red). The structural coloring reflects the B values, with blue indicating lower flexibility (lower B values) and red representing higher flexibility (higher B values). Four key regions (I-IV) with varying flexibility are annotated: Region I (residues 1-28), region II (197-222), region III (268-282), and region IV (293-312). B.1) Chain A from the SAXS model of SfC12DO exhibits regions of significant flexibility, highlighted in red, primarily regions I and **III.** B.2) Chain B follows a similar pattern but shows similarity to C12DO (PDB: 2AZQ) in region **II. C)** Root Mean Square Deviation (RMSD) values show a comparison between *P. arvilla* C12DO (PDB: 2AZQ) and SfC12DO SAXS models for chains A and **B.** The results highlight structural differences between the models, emphasizing the chain-specific deviations measured.

the lyophilization process nor in the SEC-SAXS results, suggesting that it might be a transient or a less stable form of SfC12DO or the experimental conditions used might disrupt the structural or specific conditions required for the trimer formation.

These observations underscore the substantial impact of ionic strength on the formation and stability of different oligomeric states of SfC12DO, providing valuable insights into its structural behavior in diverse environments. More importantly, it demonstrates that oligomeric transitions in SfC12DO, and potentially in other C12DOs, are more common in solution, with several oligomers maintaining their enzymatic activity. This study highlights the crucial role of experimental conditions in determining the oligomeric states of proteins and provides a deeper understanding of SfC12DO's structural dynamics. Specifically, this work examines the catalytic behavior of this C12DO under non-conventional conditions such

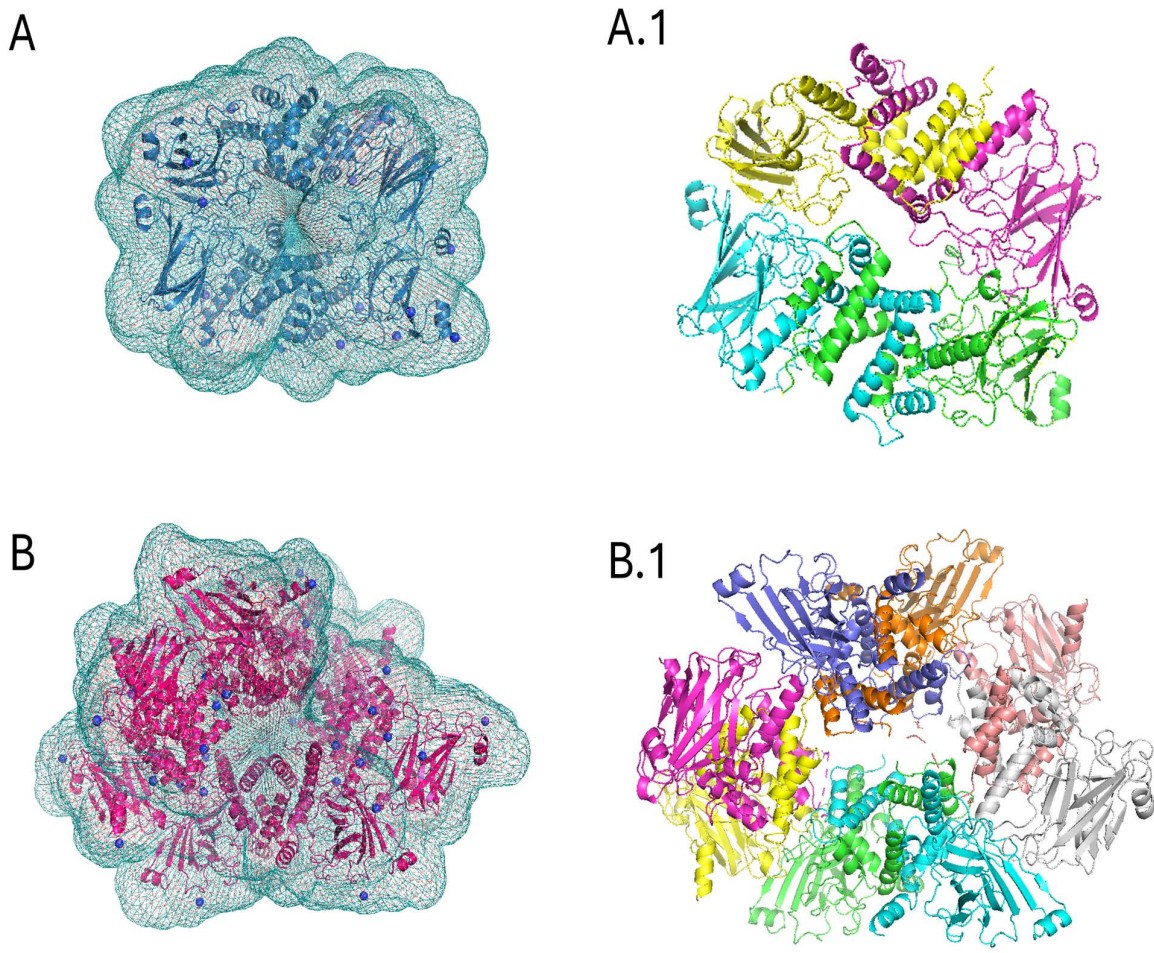

**Fig 8. SAXS models for SfC12DO tetramer and octamer. A)** SAXS envelope of the SfC12DO tetramer superimposed with its model. A.1) Cartoon representation of the tetramer, with each subunit shown in a different color **B)** SAXS envelope of the SfC12DO octamer superimposed with its model, respectively. B.1) Octamer structure with each subunit represented in a different color. Models were refined using SREFLEX and DAMMIN.

as after reconstitution of a lyophilized sample. Stressing the remarkable quaternary structure plasticity of C12DOs in specific environmental conditions and contributing to a more dynamic vision of the structural biology of proteins necessary for different processes, from basic to applied, to solve problems like, potentially, bioremediation of soils and water. They highlight the remarkable quaternary structure plasticity of SfC12DOs in specific environmental conditions, which is crucial in understanding their structural dynamics and potential implications in catalysis.

## Supporting information

**S1 Table. List of buffers.**
(PDF)

**S2 Fig. The analysis of oligomer content in lyophilized SfC12DO samples from SAXS data is presented.** The graph shows the identified oligomers in the % content of SfC12DO lyophilized samples. These oligomeric forms were identified using the OLIGOMERS tool from ATSAS.
(PDF)

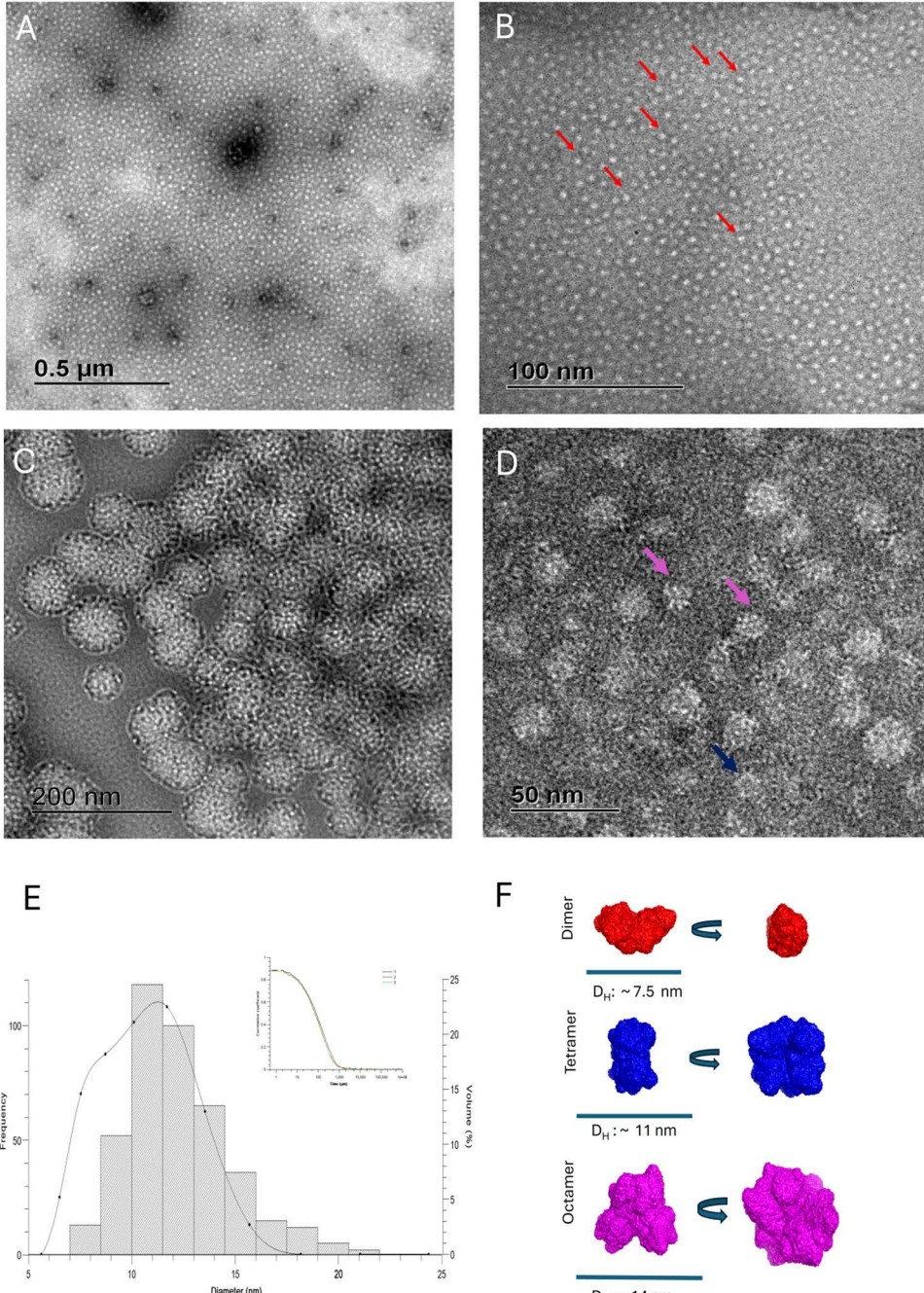

**Fig 9. Negatively stained micrographs of oligomerized SfC12DO. (A-D)** Transmission electron microscopy (TEM) images of SfC12DO oligomers at different magnifications, showing the structural organization of dimeric, tetrameric, octameric, and > 20 subunits forms **A)**. Overview of the sample with low magnification 12,000 X (scale bar: 0.5 μm). **(B)** The particles appear uniform at higher magnification, as seen at 80,000X (scale bar: 100 nm). **(C)** Aggregates of oligomers at intermediate magnification 31,500 X (scale bar: 200 nm). **(D)** Close-up of individual oligomers displaying different particle sizes and shapes corresponding to dimers (red), tetramers (blue), and octamers (pink) at 12,000 X (scale bar: 50 nm). All samples were prepared with 10 mM Tris HCl pH 8. The protein concentrations were 0.3 mg/mL, with an incubation of 5 min **(E)** Dynamic light scattering (DLS) analysis of SfC12DO oligomers, showing a histogram of particle size distribution with hydrodynamic diameters ($D_H$) corresponding to a dimer (~7.5 nm), a tetramer (~11 nm), and an octamer (~14 nm). The inset represents the autocorrelation function of the scattering intensity. **(F)** Model representation of SfC12DO oligomers based on SAXS. Dimers (red), tetramers (blue), and octamers (pink) are shown along with their respective hydrodynamic diameters **($D_H$)**.

**S3 Fig. Small-angle X-ray scattering (SAXS) analysis was conducted on non-lyophilized SfC12DO.** (A) The SAXS intensity profile is presented as a function of the scattering vector q [$\text{Å}^{-1}$]. (B) The Kratky plot, which shows $q^2I(q)$ against q, reveals the absence of a well-defined peak and a gradual increase at higher q values. This suggests that non-lyophilized SfC12DO experiences partial disorder or unfolding. (C) The Guinier plot, depicting ln(I) versus $q^2$, further indicates that non-lyophilized SfC12DO exhibits characteristics consistent with a partially unfolded or flexible conformation. (PDF)

**S4 Fig. The analysis of the distance distribution function P(r) for SfC12DO oligomers reveals the characteristics of different oligomeric states.** The curves are color-coded as follows: blue represents the dimer (p4), green denotes the tetramer (p3), red indicates the octamer (p2), and black corresponds to large aggregates (p1). The P(r) distributions exhibit distinct Rmax values, highlighting variations in size and shape among the oligomers. The distributions for the dimer and tetramer are well-defined and symmetric, while the octamer and large aggregates show broader profiles. This suggests that the latter two have increased structural complexity and potential flexibility. (PDF)

**S5 Fig. Analysis of the dimensionless Kratky plot for SfC12DO oligomers reveals a mixture of globular and flexible structures.** The curves indicate that the dimer (blue, p4) and tetramer (green, p3) display a more compact shape, while the octamer (red, p2) and larger aggregates (black, p1) show an increase at higher q values, suggesting a degree of flexibility. The variations in the profiles reflect differences in structural rigidity among the various oligomeric states. (PDF)

**S6 File. RAW data for CD, SEC, enzyme activity, and MET assays.** (XLSX)

**S7 File. SAXS data for non-lyophilized SfC12DO.** (DAT)

**S8 File. SAXS data for SfC12DO Peak 1.** (DAT)

**S9 File. SAXS data for SfC12DO Peak 2.** (DAT)

**S10 File. SAXS data for SfC12DO Peak 3.** (DAT)

**S11 File. SAXS data for SfC12DO Peak 4.** (DAT)

**S12 File. PDB model fitting to SAXS data for the dimer.** (PDB)

**S13 File. PDB model fitting to SAXS data for the tetramer.** (PDB)

**S14 File. PDB model fitting to SAXS data for the octamer.** (PDB)

**S15 File. SAXS envelope of the dimer.** (PDB)

**S16 File. SAXS envelope of the tetramer.**
(PDB)

**S17 File. SAXS envelope of the octamer.**
(PDB)

## Acknowledgments

We want to express our gratitude to B.Sc. Maricela Olvera-Rodríguez, Biol. Rosa Roman-Miranda and Ph.D. Paloma C. Gil-Rodríguez for their technical assistance. We also thank Professor Gloria Saab-Rincón's group from the Instituto de Biotecnología, UNAM, for helping with Circular Dichroism (CD) measurements. Special thanks to Professor Marcela Ayala´s Group also from the Instituto de Biotecnología, UNAM, particularly to M.Sc. Alina E. Torres-Aguirre, for kindly providing a sample of PADA-I used in the SEC-column calibration. Additionally, we acknowledge the Unidad de Microscopía Electrónica (UME) at the Instituto de Biotecnología, UNAM, for conducting the Transmission Electron Microscopy (TEM) experiments presented in this work.

## Author contributions

**Conceptualization:** Arisbeth Guadalupe Almeida-Juarez, Liliana Pardo-López, Enrique Rudiño-Piñera.

**Data curation:** Arisbeth Guadalupe Almeida-Juarez.

**Formal analysis:** Arisbeth Guadalupe Almeida-Juarez, Shirish Chodankar, Enrique Rudiño-Piñera.

**Funding acquisition:** Liliana Pardo-López, Enrique Rudiño-Piñera.

**Investigation:** Arisbeth Guadalupe Almeida-Juarez, Shirish Chodankar.

**Methodology:** Arisbeth Guadalupe Almeida-Juarez, Guadalupe Zavala-Padilla.

**Project administration:** Enrique Rudiño-Piñera.

**Supervision:** Enrique Rudiño-Piñera.

**Validation:** Arisbeth Guadalupe Almeida-Juarez, Enrique Rudiño-Piñera.

**Writing – original draft:** Arisbeth Guadalupe Almeida-Juarez, Shirish Chodankar, Liliana Pardo-López, Guadalupe Zavala-Padilla, Enrique Rudiño-Piñera.

**Writing – review & editing:** Arisbeth Guadalupe Almeida-Juarez, Enrique Rudiño-Piñera.

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
