## [Decision Letter · Decision Letter 0]

7 Jan 2025

PONE-D-24-55837Investigating the quaternary structure of a homomultimeric catechol 1,2-dioxygenase: An integrative structural biology study.PLOS ONE

Dear Dr. Rudiño-Piñera,

Thank you for submitting your manuscript to PLOS ONE. After careful consideration, we feel that it has merit but does not fully meet PLOS ONE’s publication criteria as it currently stands. Therefore, we invite you to submit a revised version of the manuscript that addresses the points raised during the review process.

Please note the both Reviewers suggested the same additional quality controls. Also, please proceed to review the minor points pointed out.

We look forward to receiving your revised manuscript.

Kind regards,

Matteo De March

Academic Editor

PLOS ONE

Journal Requirements:

[The investigation presented here was funded by the Dirección General de Asuntos del Personal Académico (DGAPA) at the Universidad Nacional Autónoma de México (UNAM) through PAPIIT grants IN226523 and IG200223. This research utilized resources from the 16-ID LIX beamline of the National Synchrotron Light Source II, a user facility managed by the U.S. Department of Energy (DOE) Office of Science, which the Brookhaven National Laboratory operates under Contract No. DE-SC0012704. Arisbeth Guadalupe Almeida-Juarez was supported by a doctoral scholarship (2019-000002-01NACF-12588) granted by CONAHCyT, Mexico, and by the incentive as a member of the program Ayudantes de Investigador nivel III provided by the SNI. Enrique Rudiño-Piñera gratefully acknowledges financial support from the institutional budget from Instituto de Biotecnología , UNAM, and the economic incentive from Sistema Nacional de Investigadoras e Investigadores (SNI), México].

6. PLOS requires an ORCID iD for the corresponding author in Editorial Manager on papers submitted after December 6th, 2016. Please ensure that you have an ORCID iD and that it is validated in Editorial Manager. To do this, go to ‘Update my Information’ (in the upper left-hand corner of the main menu), and click on the Fetch/Validate link next to the ORCID field. This will take you to the ORCID site and allow you to create a new iD or authenticate a pre-existing iD in Editorial Manager.

7. If any table files for review show as item type ‘other’ please change to item type ‘Table’ as the reviewer does not have access to these ’other’ files.

8. Please include captions for your Supporting Information files at the end of your manuscript, and update any in-text citations to match accordingly. Please see our Supporting Information guidelines for more information: http://journals.plos.org/plosone/s/supporting-information .

Reviewers' comments:

Reviewer's Responses to Questions

**Comments to the Author**

1. Is the manuscript technically sound, and do the data support the conclusions?

Reviewer #1: Yes

Reviewer #2: Yes

2. Has the statistical analysis been performed appropriately and rigorously? 

Reviewer #1: N/A

Reviewer #2: Yes

3. Have the authors made all data underlying the findings in their manuscript fully available?

Reviewer #1: No

Reviewer #2: Yes

4. Is the manuscript presented in an intelligible fashion and written in standard English?

Reviewer #1: Yes

Reviewer #2: Yes

5. Review Comments to the Author

Reviewer #1: The present study reports novel oligomeric forms of a catechol dioxygenase, an enzyme that is generally regarded as a dimeric protein.

Protein chemists are aware that proteins can form aggregates and aggregation is usually associated with (partial) unfolding, particle formation and loss of activity.

Here it is shown that before large aggregates form, a dimeric protein can form distinct tetramers and octamers, in addition to larger oligomers. The catechol dioxygenase tetramers retained full activity, demonstrating that an enzyme that retained activity after storage could nevertheless have been changed by the storage procedure. Additional quality control by light scattering or analytical gelfiltration would be advisable to control the storage procedure.

The present manuscript contains useful data, however the presentation of the results and the discussion should be improved. It remained unclear why the enzyme was lyophilized in the first place. The general problem of quality control of stored proteins should be discussed in more detail. The importance of the present study for bioremediation should also be outlined much more clearly.

Except for CD and PAGE, the results for the not-lyophilized enzyme are missing. Please add this data for gelfiltration and SAXS.

Minor issues

1. Abstract, “All these characteristics make SfC12DO a very promising candidate for extensive bioremediation applications in polluted soils or waters. “ – why? please elaborate.

2. The authors state, “All relevant data are within the manuscript and its Supporting Information files.” , however I could not find any raw data. Please add excel or csv files with raw data. Add PDB files of the models as supplementary information.

3. Difference between SfsC12DO, Sf12DO and SfC12DO?

4. page 3 line 44, the β is missing

5. Introduction: please add the alternative name 1,2-dihydroxybenzene for catechol

6. Merk → Merck

7. correction of “sample types-lyophilized-reconstituted”

8. ε = 16.8 mM, ε = 16.8 M, seems wrong and these are wrong units for an absorption coefficient (M-1 cm-1).

9. Figure 2, secondary structure cannot be predicted with ESpript, this software just displays it.

10. Figure 3, text font too small, peaks should be labelled p1, p2…, please include elution volumes of SfC12DO elution peaks in the inset (logMW vs. Ve/Vo). Please add a gelfiltration chromatogram of the enzyme which was not lyophilized for comparison. There isn’t a reference to Fig 2A in the main text, only 2B and 2C are referenced. 2C, RMSD compared to what?

11. Correction of “The CD analysis of SfC12DO's showed”

12. “non-lyophilized protein (15 ± 0.25 U/mg).” If the value is 15.00 then change to 15.00 ± 0.25 U/mg

13. Figure 5, It is unclear which panel shows which elution peak and what the diagrams actually show. If this are experimental data, where are the theoretical curves corresponding to the models? Please redo this figure and legend so that it becomes understandable for readers who are not familiar with the SAXS technique. SAXS data of the not-lyophilized enzyme would be needed.

14. Fig S3, what are the colors , what is shown here?

15. Fig. 6, there is no panel D. Panel C, I do not understand what is shown here at all.

16. “In contrast, diffusion problems might prevent it” – it = the substrate?

Reviewer #2: The authors describe the multimeric states of a catechol dehydrogenase enzyme and its impact on its activity. Overall, the manuscript is well laid out. I have a few recommendations for the authors and a few clarifications for the methods.

1. in Figure 2C, the authors have a table explaining the structural nuances for the few studied C12DOs; this would be improved for the reader if they could add the structures and their comparisons for domains and catalytic sites that would convey much more effectively.

2. What is the buffer for the enzymatic activity assay? Does the assay performance change for the Lyo and non-lyo material in different salinity?

3. For Figure 3B, can the authors add both the native and denaturing gel bands? Also, why not have all four peaks on the gel? They're tentatively in the right MW size range to run a native gel for all four peaks.

4. There definitely needs to be an overlay of the SEC profile for the non-lyo and lyo material to make that clear these multimers reflect the nature of the material.

5. IF there are gel lanes for all four peaks, then the enzymatic assay from those fractions is a good continuation.

I think the authors making these edits will improve the overall manuscript.

6. PLOS authors have the option to publish the peer review history of their article (what does this mean? ). If published, this will include your full peer review and any attached files.

**Do you want your identity to be public for this peer review?** For information about this choice, including consent withdrawal, please see our Privacy Policy .

Reviewer #1: No

Reviewer #2: No

---

## [Author Response · Author response to Decision Letter 0]

21 Feb 2025

PONE-D-24-55837

Investigating the quaternary structure of a homomultimeric catechol 1,2-dioxygenase: An integrative structural biology study.

PLOS ONE

Dear Dr. Rudiño-Piñera,

Thank you for submitting your manuscript to PLOS ONE. After careful consideration, we feel that it has merit but does not fully meet PLOS ONE’s publication criteria as it currently stands. Therefore, we invite you to submit a revised version of the manuscript that addresses the points raised during the review process.

Please note the both Reviewers suggested the same additional quality controls. Also, please proceed to review the minor points pointed out.

● A rebuttal letter that responds to each point raised by the academic editor and reviewer(s). You should upload this letter as a separate file labeled 'Response to Reviewers'.

● A marked-up copy of your manuscript that highlights changes made to the original version. You should upload this as a separate file labeled 'Revised Manuscript with Track Changes'.

● An unmarked version of your revised paper without tracked changes. You should upload this as a separate file labeled 'Manuscript'.

We look forward to receiving your revised manuscript.

Kind regards,

Matteo De March

Academic Editor

PLOS ONE

Journal Requirements:

[The investigation presented here was funded by the Dirección General de Asuntos del Personal Académico (DGAPA) at the Universidad Nacional Autónoma de México (UNAM) through PAPIIT grants IN226523 and IG200223. This research utilized resources from the 16-ID LIX beamline of the National Synchrotron Light Source II, a user facility managed by the U.S. Department of Energy (DOE) Office of Science, which the Brookhaven National Laboratory operates under Contract No. DE-SC0012704. Arisbeth Guadalupe Almeida-Juarez was supported by a doctoral scholarship (2019-000002-01NACF-12588) granted by CONAHCyT, Mexico, and by the incentive as a member of the program Ayudantes de Investigador nivel III provided by the SNI. Enrique Rudiño-Piñera gratefully acknowledges financial support from the institutional budget from Instituto de Biotecnología , UNAM, and the economic incentive from Sistema Nacional de Investigadoras e Investigadores (SNI), México].

ANSWER to comment 3: We examined the discrepancies between the "Funding Information" and the "Financial Disclosure" sections and confirmed that both are consistent with each other. Additionally, we have added the following sentence to the end of the new cover letter: “Finally, we declare that the funders had no role in the study design, data collection, data analysis, decision to publish, or preparation of the manuscript submitted to PLOS One.”

ANSWER to comment 4: We confirm that Professor Enrique Rudiño-Piñera (enrique.rudino@ibt.unam.mx), the corresponding author of this manuscript, is affiliated with the Instituto de Biotecnología at the National Autonomous University of Mexico, which has an active agreement with PLOSOne.

ANSWER to comment 5: We have updated our original data availability statement to indicate that the data will be available upon acceptance. We are also uploading an Excel file labeled "RAW DATA," which contains the original data recorded for all the experiments described in the manuscript submitted to PLOSone.

6. PLOS requires an ORCID iD for the corresponding author in Editorial Manager on papers submitted after December 6th, 2016. Please ensure that you have an ORCID iD and that it is validated in Editorial Manager. To do this, go to ‘Update my Information’ (in the upper left-hand corner of the main menu), and click on the Fetch/Validate link next to the ORCID field. This will take you to the ORCID site and allow you to create a new iD or authenticate a pre-existing iD in Editorial Manager.

Answer to comment 6: Thank you and we are sorry for missing the required information. The ORCID ids for all authors are:

Enrique Rudiño- Piñera: 0000-0001-8170-6379

Arisbeth Guadalupe Almeida-Juarez: 0000-0001-9963-9151

Liliana Pardo-López: 0000-0002-8927-1733

Shirish Chodankar: 0000-0003-4850-2926

Guadalupe Zavala: 0000-0001-8489-9823

7. If any table files for review show as item type ‘other’ please change to item type ‘Table’ as the reviewer does not have access to these ’other’ files.

Answer to comment 7: Thank you for the comment. We confirm that the only table submitted in this manuscript is not labeled as others.

Answer to comment 8: Thank you for the comment. We confirm that the figures in the supplementary material have captions.

Reviewers' comments:

Reviewer's Responses to Questions

Comments to the Author

1. Is the manuscript technically sound, and do the data support the conclusions?

Reviewer #1: Yes

Reviewer #2: Yes

2. Has the statistical analysis been performed appropriately and rigorously?

Reviewer #1: N/A

Reviewer #2: Yes

3. Have the authors made all data underlying the findings in their manuscript fully available?

Reviewer #1: No

Reviewer #2: Yes

4. Is the manuscript presented in an intelligible fashion and written in standard English?

Reviewer #1: Yes

Reviewer #2: Yes

5. Review Comments to the Author

Reviewer #1: The present study reports novel oligomeric forms of a catechol dioxygenase, an enzyme that is generally regarded as a dimeric protein.

Protein chemists are aware that proteins can form aggregates and aggregation is usually associated with (partial) unfolding, particle formation and loss of activity.

Here it is shown that before large aggregates form, a dimeric protein can form distinct tetramers and octamers, in addition to larger oligomers. The catechol dioxygenase tetramers retained full activity, demonstrating that an enzyme that retained activity after storage could nevertheless have been changed by the storage procedure. Additional quality control by light scattering or analytical gelfiltration would be advisable to control the storage procedure.

The present manuscript contains useful data, however the presentation of the results and the discussion should be improved. It remained unclear why the enzyme was lyophilized in the first place. The general problem of quality control of stored proteins should be discussed in more detail. The importance of the present study for bioremediation should also be outlined much more clearly.

Except for CD and PAGE, the results for the not-lyophilized enzyme are missing. Please add this data for gel filtration and SAXS.

Thank you for your observation regarding the use of a lyophilized version of SfC12DO. To clarify, we have added the following lines in the revised manuscript:

Page 12, line 231-236: “In a preliminary characterization of SfC12DO oligomers, freshly purified SfC12DO was subjected to an initial SAXS (static mode) measurement. However, the samples exhibited signs of protein unfolding (S2 Fig.). During shipping to the synchrotron facility, the protein was unable to withstand the transport conditions and precipitated. To mitigate aggregation and enhance stability during shipping and long-term storage, we investigated lyophilization as a preservation strategy”.

Furthermore, regarding the information that was previously missing about the non-lyophilized SfC12DO, this data is now presented in the new Figure 4 and in Supplementary Figure S3.

We acknowledge that we did not provide any arguments regarding the protein stability or perform quality control on the lyophilized SfC12DO. The main reason for this omission is that our study focused on demonstrating that the enzyme, despite changing its quaternary structure, retains its enzymatic activity in the reconstituted sample. We agree that if our goal had been to prove the applicability of this lyophilized enzyme in bioremediation, additional experiments and controls would have been necessary. However, this objective is beyond the scope of the present study.

Minor issues

1. Abstract, “All these characteristics make SfC12DO a very promising candidate for extensive bioremediation applications in polluted soils or waters. “ – why? please elaborate.

Answer to comment 1: First, we acknowledge that our claim stating that SfC12DO is a “very promising candidate for extensive bioremediation applications” is based solely on its ability to maintain enzymatic activity in a reconstituted sample that was previously lyophilized. We agree that to support such a strong statement, additional experiments and controls must be conducted beyond the results presented in this work. Consequently, the following sections of the manuscript have been revised:

In the abstract the final new sentencein Page 3, line 38-39 is: “All these characteristics make SfC12DO a putative candidate for bioremediation applications in polluted soils or waters.”

Page 25, line 495-496: “...to solve problems like, potentially, bioremediation of soils and water”

Page 25, line 499: The sentence “and its potential implications in bioremediation” were erased.

2. The authors state, “All relevant data are within the manuscript and its Supporting Information files.” , however I could not find any raw data. Please add excel or csv files with raw data. Add PDB files of the models as supplementary information.

Answer to comment 2: You are correct; we did not include the raw data files in the original submission. Along with the corrected version, we are uploading an Excel file containing the raw data for figures 1 to 9 and supplementary figures S2 to S5, also protein models and SAXS data were added.

3. Difference between SfsC12DO, Sf12DO and SfC12DO?

Answer to comment 3. Thank you for your observation. There is no distinction between SfsC12DO, Sf12DO, and SfC12DO; these variations stem from an inconsistency in the use of acronyms in the original text. All terms refer to the same protein, SfC12DO, as clarified in the revised version of the manuscript. We apologize for the error.

4. page 3 line 44, the β is missing

Answer to comment 4. Thank you for pointing out this mistake; you are correct. We have corrected the text, and the missing "β" has been added on page 3, line 44 of the revised manuscript. We appreciate your attention to detail.

5. Introduction: please add the alternative name 1,2-dihydroxybenzene for catechol

Answer to comment 5. Thank you for your suggestion. We have updated the introduction to include the alternative name "1,2-dihydroxybenzene" for catechol, enhancing clarity and avoiding misinterpretations on page 3, line 45 of the revi

---

## [Decision Letter · Decision Letter 1]

30 Mar 2025

Investigating the quaternary structure of a homomultimeric catechol 1,2-dioxygenase: An integrative structural biology study.

PONE-D-24-55837R1

Dear Prof. Enrique Rudiño-Piñera,

We’re pleased to inform you that your manuscript has been judged scientifically suitable for publication and will be formally accepted for publication once it meets all outstanding technical requirements.

Kind regards,

Matteo De March

Academic Editor

PLOS ONE

Additional Editor Comments (optional):

Reviewers' comments:

Reviewer's Responses to Questions

**Comments to the Author**

1. If the authors have adequately addressed your comments raised in a previous round of review and you feel that this manuscript is now acceptable for publication, you may indicate that here to bypass the “Comments to the Author” section, enter your conflict of interest statement in the “Confidential to Editor” section, and submit your "Accept" recommendation.

Reviewer #1: All comments have been addressed

2. Is the manuscript technically sound, and do the data support the conclusions?

Reviewer #1: Yes

3. Has the statistical analysis been performed appropriately and rigorously? 

Reviewer #1: Yes

4. Have the authors made all data underlying the findings in their manuscript fully available?

Reviewer #1: Yes

5. Is the manuscript presented in an intelligible fashion and written in standard English?

Reviewer #1: Yes

6. Review Comments to the Author

Reviewer #1: The authors have significantly improved the manuscript according to the referee reports. I would recommend to publish the paper now.

7. PLOS authors have the option to publish the peer review history of their article (what does this mean? ). If published, this will include your full peer review and any attached files.

**Do you want your identity to be public for this peer review?** For information about this choice, including consent withdrawal, please see our Privacy Policy .

Reviewer #1: No

---

## [Editor Report · Acceptance letter]

PONE-D-24-55837R1

PLOS ONE

Dear Dr. Rudiño-Piñera,

I'm pleased to inform you that your manuscript has been deemed suitable for publication in PLOS ONE. Congratulations! Your manuscript is now being handed over to our production team.

Kind regards,

on behalf of

Dr. Matteo De March

Academic Editor

PLOS ONE